# Do Ecosystem Service Value Increase and Environmental Quality Improve due to Large–Scale Ecological Water Conveyance in an Arid Region of China?

**Xiyi Wang [1,2,3], Shuzhen Peng [1], Hongbo Ling [2,3,*], Hailiang Xu [2,3] and Tingting Ma [1]**

[1]   Department Taishan University, Taian 271000, Shandong, China; wangxiyi@tsu.edu.cn (X.W.);
      pengshuzhen@tsu.edu.cn (S.P.); matingting@tsu.edu.cn (T.M.)
[2]   Xinjiang Institute of Ecology and Geography, Chinese Academy of Sciences, Urumqi 830011, Xinjiang, China;
      xuhl@ms.xjb.ac.cn
[3]   State Key Laboratory of Desert and Oasis Ecology, Xinjiang Institute of Ecology and Geography, Chinese
      Academy of Sciences (CAS), Urumqi 830011, China
*   Correspondence: linghb@ms.xjb.ac.cn

**Abstract:** With the rapid development of the economy and the intensification of human activities, ecological systems have been degraded, especially in arid areas. The lower reaches of the Tarim River represent a typical arid area in China. Since 2000, the Chinese government has been heavily investing in the protection and restoration of the natural ecological environment of the lower reaches of the Tarim River. In this study, we aimed to resolve two key scientific issues: (1) reveal the changing characteristics of land-use in the region and identify the changes in ecosystem service value caused by these land-use changes and (2) evaluate whether the environmental quality has improved or worsened. The objective of this study is to verify whether the ecological water conveyance project promotes an increase in the ecosystem service value, with an improvement in the ecological environment, to thereby provide references for the evaluated effects of ecological water conveyance for the management of water resources. In this way, economic development can support environmental protection. Thus, the economy can be sustainably developed. Hence, based on the remote sensing data of land-use in 1990, 2000, 2010, and 2016, with the value coefficients proposed by Constanza in 1997 and changing characteristics in the land-use, the ecological service value, and environmental conditions from 1990 to 2016 were analysed. According to our results, from 1990 to 2016, the ecosystem service value has increased substantially, indicating that the benefits of ecological water conveyance were significant. The environmental condition index increased by 21.14%, showing that the ecological environment has improved. However, the environmental quality remained low. In the future, we should formulate plans for reasonable land-use that control the replacement of woodlands and grasslands with farmlands and construction. The results of this study provide a scientific basis and practical guide for restoring inland river ecosystems in arid regions.

**Keywords:** ecosystem service value; land-use; environmental quality; ecological water conveyance; lower reaches of Tarim river; arid area

## 1. Introduction

The change in the ecosystem service value, which is an important part of the "IHDP" (International Human Dimensions Programme on Global Environmental Change), is one of the core aspects of global ecological environment change research [1,2]. Ecosystems are mainly affected by economic,

technological, social and political culture, so this change has been followed with interest by scholars worldwide [3,4]. Quantifying the service value of different ecosystems with economic value indicators, thereby accounting for the relationship between natural ecosystems and human activities, is significant for the exploitation and utilization of natural resources [5–7]. With the development of the economy, resource shortages and environmental pollution have become common problems that influence the sustainable development of human society. Research on environmental quality has also attracted widespread attention [8–10]. Analysing the changes in ecosystem service values and environmental quality can provide a theoretical basis for the utilization and allocation of resources, and therefore, such analyses are significant for the sustainable development of the economy.

In 1977, Westman [11] proposed the concept of 'nature's services' and the problems related to assessing the value of these services. However, at that time, accurately measuring the value of most services provided by earth's ecosystems was difficult, and corresponding theories and methodologies for value assessment were lacking. Therefore, research progress in this area has been slow. Daily and Constanza began by providing an estimate of the value of the global biosphere ecosystem service prior to 1997 [12,13]. They divided the global biosphere into 16 ecosystem types and divided the ecosystem services into 17 types. In the same year, Pimentel et al. [14] conducted a comparative study of the world's biodiversity and the economic value of American biodiversity. Since the beginning of the 21st century, the intensity of human development and utilization of ecological resources has been increasing, resulting in environmental pollution and destruction [15,16], which lead to problems with environmental quality assessment. Hence, 'sustainable development' has become the theme of harmonious development between humans and nature. How to balance the relationships between economic development and environmental protection and to avoid serious damage to the ecological environment caused by the traditional development model of 'first development, then protection', is the main problem at present. Research related to ecological protection and estimation of gross ecosystem product (GEP) has been developing rapidly [17–19]. At present, many methods exist for assessing ecosystem service values and environmental quality [20–22], and the results all have been digitized. They have strong operability and applicability and have been applied widely [23–25].

In the past few decades, China has pursued economic developments and has become the second-largest economy in the world [26]. Since beginning to 'reform and open', China has lifted hundreds of millions of people out of poverty. However, the rapid development of the economy has led to serious environmental degradation. Along the Yangtze River, due to the extensive logging and erosion caused by forestry activities, a serious flood occurred in 1998. Hence, the Chinese government has invested much manpower and material resources to strengthen environmental protection and promote ecological restoration. The Tarim River, which is in an arid region, is also an important river in China, and the ecological environment of the river basin is very fragile. In 2000, the Chinese government invested 10 billion 700 million yuan to conduct a comprehensive management project for the Tarim River Basin. This project aimed to restore the damaged ecological environment in the lower reaches of the Tarim River, which has attracted worldwide attention [27].

At present, some studies are underway to study the concept of ecological water conveyance. Some experts believe an ecological water conveyance project will protect and restore the ecosystem with natural vegetation [28]. In addition, some experts believe that an ecological water conveyance is a project that will balance water resources without destroying the ecological environment, with the project being integrated with nature [29]. At present, most of the research on ecological water conveyance is focused on China's arid areas, especially those of the Heihe and Tarim River basins [30,31]. The research on ecological water conveyance in the Heihe River basin mainly focuses on the influence of the water conveyance on the vegetation [32,33], the characteristics of the water resources in the basin [34], and the comprehensive management of the water eco–economic system [35]. Research on the lower reaches of the Tarim River mainly focuses on the impacts of the ecological water conveyance on the distribution, structural characteristics, physiological and ecological characteristics of the vegetation [36–39]. Some scattered cases studying ecosystem service value and environmental quality in the lower reaches of the

Tarim River exist [40,41]. However, these studies still focus on the initial stage, and research on the ecosystem service value and environmental quality of this region is lacking.

The lower reaches of the Tarim River are typical inland river basins in arid regions of China, which means that water resources are very valuable in this region. This fact leads to several important problems: Is it wise to use such huge water resources for ecological restoration? How does the land cover change during the process of ecological water conveyance? How has the service value of the entire ecosystem changed? Due to the ecological water conveyance, the economy in the lower reaches of the Tarim River has also been developed. However, some studies have found that environmental quality deteriorates in the early stage of economic development but improves with economic growth when the economy reaches a certain stage [42]. Therefore, how was the environmental quality changed in the lower reaches of the Tarim River? All of the above questions indicate urgent problems to be solved. Based on the above issues, the latest data in 2016 were extracted, and remote sensing data in 1990, 2000, 2010, and 2016 were collected. Corresponding methods were used to calculate values from these data. The objectives of this study are to (1) determine the evaluation coefficients of ecological values in the lower reaches of the Tarim River and explore whether these coefficients can be used to evaluate changes in local ecosystem service values, (2) assess the impacts of land-use change on the ecosystem service value during the process of ecological water conveyance, and (3) analyse the changing trend in the ecological environmental quality due to ecological water conveyance.

## 2. Data and Methods

### 2.1. Study Area Description

The lower reaches of the Tarim River are located in Yuli County, Xinjiang. East of this region is the Kuruk Desert, and to the west is the Taklimakan Desert. The lower reaches of the Tarim River provide an important channel from Xinjiang to the mainland of China through Qinghai Province (Figure 1). The region belongs to a warm temperate zone, and the climate is continental arid. The annual average precipitation is 17.4–42.0 mm, whereas the annual average potential evaporation is as high as 2500–3000 mm [43]. Furthermore, much sand and wind are present, and the maximum wind speed can reach 40 m·s$^{-1}$. To save the strategic 'green corridor' in the lower reaches of the Tarim River, an emergency ecological water conveyance project in the lower reaches of the Tarim River began in May 2000. The aim of the project was to raise the water level on both sides of the river and save the declining natural vegetation [44,45]. Since the implementation of the ecological water conveyance project, vegetation on both sides of the river has been supplied by river water and groundwater. From 2000 to 2016, 17 ecological water transfers were carried out in the lower reaches of the Tarim River, with a cumulative discharge of $58.14 \times 10^8$ m$^3$ and an average annual discharge of $3.42 \times 10^8$ m$^3$. The water head reached Taitema Lake 14 times, which effectively replenished the groundwater. With the rise in the groundwater level, the water quality of the groundwater and the water environment improved obviously. From 2000 to 2016, the groundwater depth increased from 9.8–10.1 m to 2.1–5.3 m at 1 km away from the river. The area of vegetation restoration and improvement reached 2285 km$^2$, and the area of new vegetation coverage reached 362 km$^2$. The area of sand decreased by 854 km$^2$, and a wetland, approximately 223 km$^2$, formed around Taitema Lake. Large areas of dead or dying vegetation have recovered. Vegetation coverage and species diversity have rebounded [46]. The species include 3 types: Trees, shrubs, and grass. These plant species, constituting the desert riparian forest community, play a leading role in ecosystem structure, function and vegetation landscape. Land types in this area, among other aspects, include farmland, woodland, and grassland. With the implementation of an ecological water conveyance, the structural characteristics of the land types have changed significantly.

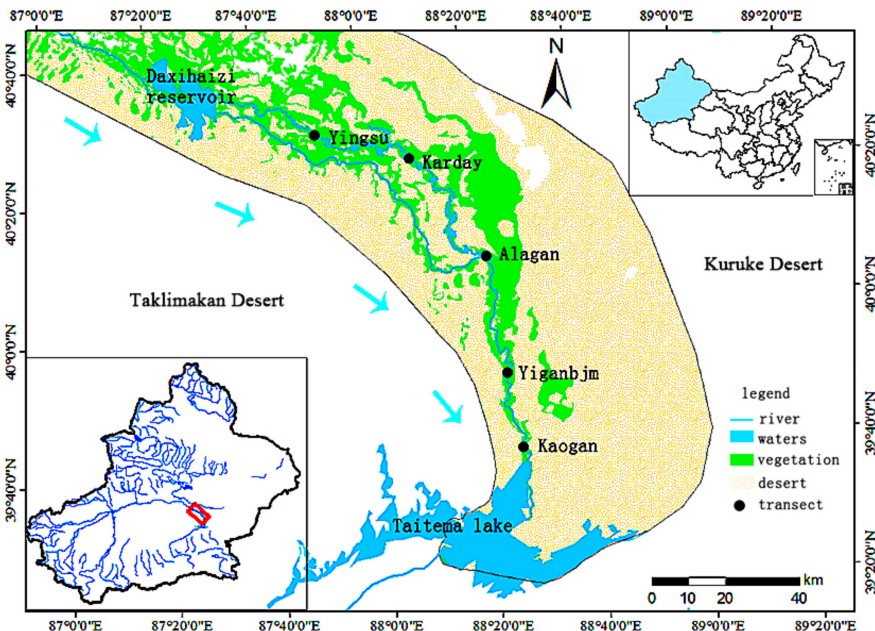

**Figure 1.** Sketch map of the low reaches of Tarim River.

## 2.2. Methods

### 2.2.1. Remote Sensing Image Interpretation and Classification

The basic data were Landsat-TM (Thematic Mapper) image data and CBERS (China–Brazil Earth Resource Satellite) data from the lower reaches of the Tarim River. Images were obtained in August 1990, 2000, 2010, and 2016 during periods of abundant water in the basin, with large amounts of water in the rivers and reservoirs. Furthermore, natural vegetation and crops were growing luxuriantly, and land features were obvious. Thus, we obtained four images. First, Erdas software was used to provide geometric correction and registration of the four remotely sensed images, whereas, for visual interpretation and digitization of the images, the ArcInfo module supported by ArcGIS 10 was used. Second, the interpretation of the river basin was revised by on-the-spot investigation and calibration, and the accuracy of the data had to exceed 80% to meet the needs of the study. Third, according to the final account of state revenue and expenditure, and the needs of the survey, the land-use pattern of the survey area was classified. According to the resources and utilization attributes, land in the lower reaches of the Tarim River was divided into six categories: Farmland, Woodland, Grassland, Water bodies, Construction land, and Unused land. Land-use in different years is shown in Figure 2.

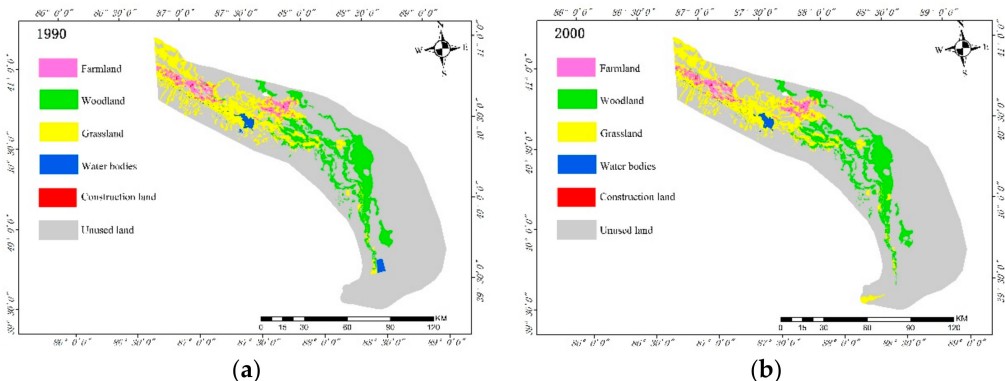

(**a**)          (**b**)

**Figure 2.** *Cont.*

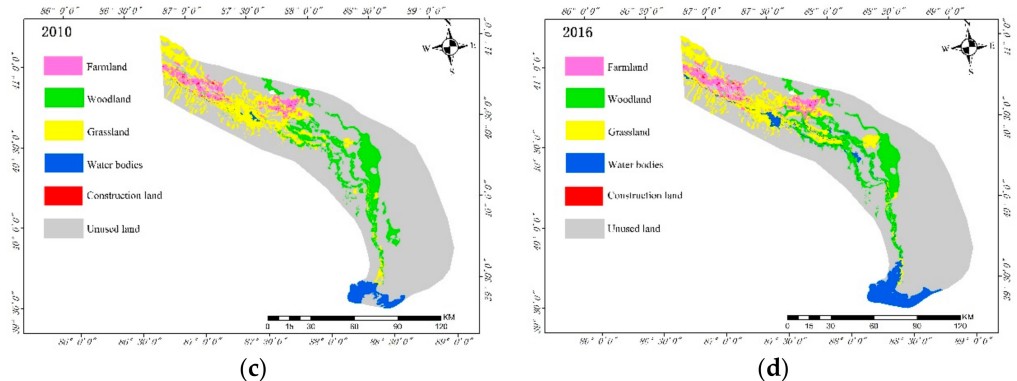

**Figure 2.** Land-use in the lower reaches of the Tarim River in four years from 1990 to 2016. (**a**) was the land-use in the lower reaches of the Tarim River in 1990. (**b**) was the land-use in the lower reaches of the Tarim River in 2000. (**c**) was the land-use in the lower reaches of the Tarim River in 2010. (**d**) was the land-use in the lower reaches of the Tarim River in 2016.

In addition, data on the river length, reservoir area, and water resources were provided by the Tarim River Basin Authority.

### 2.2.2. Analysis Methods of Land Coverage

In this article, the characteristics of land cover change are described according to the single land type change rate ($R_k$) and land-use change rate ($R$). If the study area includes $n$ types of land-use, the dynamics (the rate of changes in the quantity of a certain type of land-use within a certain period of time) of type $k$ can be calculated by Equation (1):

$$R_k = \frac{|U_{bk} - U_{ak}|}{U_{ak}} \tag{1}$$

where $k = 1,2,3, \ldots , n$, and $U_{ak}$ and $U_{bk}$ represent the area of type $k$ at the beginning and end of the study, respectively (km$^2$).

### 2.2.3. Methods for Evaluating Ecosystem Service Values

At present, the study of value coefficients includes a value coefficient proposed by Constanza et al. [13]. In addition, the "equivalent factor table of China's terrestrial ecosystem services value" put forward by 200 ecologists was summarized by Xie et al. [47]. However, Shi et al. [48] compared several value coefficients and noted that the value system of Constanza et al. (Table 1) was in line with the actual situation of arid areas. That is, it can highlight the value of water in arid areas, and the desert system is assigned 0, which is consistent with the actual situation of arid areas. In arid areas, water is an important factor that determines the characteristics of the ecosystem and determines the two ecological environment evolution processes of oasization and desertification, which are extremely contradictory and conflicting. The role and value of water should be highlighted in arid areas. Thus, the value system of Constanza et al. is more applicable to the actual situation of arid areas. In addition, under the value system of Constanza et al., unused land includes desert and Gobi Desert. Desert and Gobi Desert, although not heavily weighted, cover huge areas and account for a large proportion of the arid areas. Therefore, the value of unused services is 0, which is more in line with the actual situation of arid areas. As a result, this article chooses the value system of Constanza et al. as the basis of this study.

**Table 1.** Value equivalent of unit area service of an ecosystem [13].

| Service Function | Farmland | Woodland | Grassland | Water Bodies | Construction Land | Unused Land |
|---|---|---|---|---|---|---|
| Food Production | 1.00 | 0.80 | 1.24 | 4.74 | 0.76 | 0.00 |
| Raw Material Production | 0.00 | 2.56 | 0.00 | 1.96 | 0.00 | 0.00 |
| Gas Regulation | 0.00 | 0.00 | 0.13 | 2.46 | 0.00 | 0.00 |
| Climate Regulation | 0.00 | 2.65 | 0.00 | 0.08 | 0.00 | 0.00 |
| Hydrologic Regulation | 0.00 | 0.09 | 0.06 | 0.35 | 0.14 | 0.00 |
| Waste Disposal | 0.00 | 1.61 | 1.61 | 0.08 | 12.31 | 0.00 |
| Soil Conservation | 0.00 | 8.65 | 0.56 | 0.00 | 0.00 | 0.00 |
| Maintenance of Biodiversity | 0.70 | 0.33 | 0.89 | 5.63 | 0.00 | 0.00 |
| Provide Aesthetic Landscape | 0.00 | 1.26 | 0.04 | 26.94 | 4.26 | 0.00 |
| Total | 1.70 | 17.95 | 4.53 | 42.24 | 17.47 | 0.00 |

Therefore, while taking the value coefficients of Constanza et al. as the basis, this study calculates the ecosystem service value in the area. The value coefficients of Constanza et al. consider cognitive factors with subjective values, and the willingness to pay reflects this subjective value cognition. People's cognition of ecosystem service value is changing along with the development of the social economy, so stage coefficients of social development (*l*) can be used to show responses to the willingness to pay for ecosystem services [49]. The formula of the coefficients of the social development stage is as follows:

$$l = \frac{1}{1 + e^{(3-1/E_n)}} \tag{2}$$

where *l* is the stage coefficient of social development, and $E_n$ is the Engel coefficient. In this study, $E_n$ was assumed to be the Engel coefficient of Yuli County in Xinjiang (the lower reaches of the Tarim River are close to Yuli County).

The coefficients of the social development stage can reflect people's overall willingness to pay for ecosystem services, but distinguishing the preferences for productive functions (food production and raw material production) and service functions (gas regulation, climate regulation, hydrological regulation, waste disposal, soil conservation, biodiversity maintenance, and aesthetic landscape provision) is impossible. Therefore, this article introduces the research results of Shi et al. [50], whereby *T* is the correction factor to distinguish the preferences of different service functions based on the stage coefficient of social development. The calculation formulas were as follows:

$$T_p = l_r \tag{3}$$

$$T_s = \frac{l_r}{l_c} \tag{4}$$

$T_p$ and $T_s$ are the coefficients of productive function and service function respectively. $l_r$ and $l_c$ are the stage coefficients of the social development of the study area and China respectively.

In summary, the revised model of ecosystem service value coefficient is proposed:

$$e_c = eT \tag{5}$$

$e_c$ is the revised value coefficient, *e* is a value coefficient of Constanza et al. [13], *T* is a correction factor.

The ecological service value of the study area was estimated by the estimation method of Constanza et al. [13], as:

$$ESV = \sum_{i=1}^{6}\sum_{j=1}^{9} e_{ij}S_iV \tag{6}$$

*ESV* is the total value of ecosystem services in the study area (dollar), $e_{ij}$ is the equivalent value of ecosystem service function of *j* belong to land *I* (dollar), $S_i$ is the area of land *i* (hm$^2$), *V* is the economic value of unit farm ecosystem to provide food production services. *V* can be calculated as [47].

$$V = \frac{1}{7}\sum_{i=1}^{n} \frac{m_ip_iq_i}{M} \tag{7}$$

In equation (7), *i* is the species of crop, the main crops in the lower reaches of Tarim River are wheat and corn, $p_i$ is the prices of respective years of grain crops (dollar /t), $q_i$ is the yield per unit area (t/hm$^2$), $m_i$ is the area of grain crop *i* (hm$^2$), *M* is the total area of grain crop (hm$^2$). According to the "Xinjiang Statistical Yearbook" and "Xinjiang Production and Construction Corps Yearbook", *V* was calculated as 240.53 dollar/(hm$^2$·a).

In this study, a sensitivity analysis was used to examine the degree to which the accuracy of the value coefficient influences the calculated results. By adjusting the corresponding ecological service value coefficients of farmland, woodland, grassland, water bodies, construction land, and unused land, sensitivity was used for the analysis. The formula was as follows:

$$CS = \left| \frac{\left(ESV_j - ESV_i\right)/ESV_i}{\left(e_{jk} - e_{ik}\right)/e_{ik}} \right| \tag{8}$$

The *ESV* is the total service value (dollars), *e* is the value coefficient, *i* and *j* are the value coefficients before and after adjustment, *k* is the type of land-use (km$^2$), and *CS* is sensitivity, which refers to the change in the *ESV* caused by the 1% change of *e*. *CS* > 1 indicates *ESV* was resilient to *e*, and *CS* < 1 indicates *ESV* lacks flexibility to *e*. The larger the ratio is, the more critical the accuracy of *e*.

### 2.3. Calculation of the Ecological Environment Index (EI)

To evaluate the ecological environment changes in the basin, characteristics of ecological environment change were analysed by the methods of the ecological environment index published by the "State Environmental Protection Administration of China". The index system of the ecological environment quality evaluation and calculation method of each index refers to the standard "Technical specification for ecological environmental assessment (Trial Implementation)" [51].

EI (environment index) = 0.25 × Bioabundance index + 0.2 × Vegetation coverage index + 0.2 × Water network density index + 0.2 × (100-Land degradation index) + 0.15 × Environmental quality index.

Bioabundance index = $A_{bio}$ × (0.35 × woodland + 0.21 × grassland + 0.28 × water bodies + 0.11 × farmland + 0.04 × construction land + 0.01 × unused land)/area, $A_{bio}$: The normalization coefficient of Bioabundance index.

Vegetation coverage index= $A_{veg}$ × (0.38 × woodland + 0.34 × grassland + 0.19 × farmland + 0.07 × construction land + 0.02 × unused land)/area, $A_{veg}$: Normalization coefficient of Vegetation coverage index.

Water network density index = $A_{riv}$ × river length/area + $A_{lak}$ × Lake (Offshore) area/area + $A_{res}$×water resource amount/area, $A_{riv}$: Normalization coefficient of river length, $A_{lak}$: Normalization coefficient of the lake area, $A_{res}$: Normalization coefficient of water resource amount.

Land degradation index = $A_{ero}$ × (0.05 × area of lightly erosion + 0.25 × area of moderate erosion + 0.7 × area of severe erosion)/area, $A_{ero}$: Normalization coefficient of land degradation index.

Environmental quality index = $0.4 \times (100\text{-}As_{o2} \times$ emission of $SO_2$/area) + $0.4 \times (100\text{-}A_{COD} \times$ emission of COD/average annual rainfall of region) + $0.2 \times (100\text{-}As_{o1} \times$ emission of solid waste/area), $A_{SO2}$: Normalization index of $SO_2$, $As_{o1}$: Normalization index of solid waste, $A_{COD}$: Normalization index of COD.

The empirical data of the normalization index recommended by China's Ministry of Environmental Protection are indicated in Table 2.

**Table 2.** Normalization coefficient of each subitem.

| Normalization Constant | Numerical Value |
|---|---|
| Biological abundance index | 400.62 |
| Vegetation cover index | 355.24 |
| River length | 46.63 |
| Lake area | 17.88 |
| Water resources | 61.42 |
| Land degradation index | 146.33 |
| $SO_2$ | 0.06 |
| COD (Chemical Oxygen Demand) | 0.33 |
| Solid waste | 0.77 |

## 3. Results and Analysis

### 3.1. Changes in Land-Use Types Before and After Ecological Water Conveyance

According to the characteristics of the land classification and ecological service value scale of the study area, this study regards arable land as farmland. In addition, woodland includes forestland, sparse woodland, shrubs and so on. Grassland includes high–coverage grassland, moderate–coverage grassland and low–coverage grassland. Water bodies include rivers, lakes, marshes, etc. Construction land includes residential areas, factories, farms, etc. Unused land includes sandy land, saline-alkali soil, bare land, bare rock gravel, and other unused lands. Based on the interpretation of the analysed remote sensing and field survey data, variation characteristics of six different types of land areas were obtained in different years (Tables 3 and 4).

**Table 3.** Areas of different types of land areas in different years.

| Type of Land-Use | 1990 | 2000 | 2010 | 2016 |
|---|---|---|---|---|
| Farmland | 337.39 | 355.02 | 458.38 | 510.50 |
| Woodland | 1493.93 | 1404.67 | 1350.29 | 1280.89 |
| Grassland | 1704.77 | 1704.84 | 557.36 | 1568.18 |
| Water bodies | 119.23 | 72.44 | 1404.53 | 410.83 |
| Constrution land | 13.86 | 14.84 | 15.29 | 17.71 |
| Unused land | 9828.96 | 9946.35 | 9718.2 | 9715.84 |

**Table 4.** Change characteristics of the area of different land-use types from 1990 to 2016.

| Type of Land-Use | 1990–2000 | | 2000–2010 | | 2010–2016 | | 1990–2016 | |
|---|---|---|---|---|---|---|---|---|
| | Variation of Area (km²) | Change Rate (%) | Variation of Area (km²) | Change Rate (%) | Variation of Area (km²) | Change Rate (%) | Variation of Area (km²) | Change Rate (%) |
| Farmland | 17.63 | 5.23 | 103.36 | 29.11 | 52.12 | 11.37 | 173.11 | 51.31 |
| Woodland | −89.25 | −5.97 | −54.38 | −3.87 | −69.40 | −5.14 | −213.03 | −14.26 |
| Grassland | 0.07 | 0.004 | −1147.48 | −67.31 | 1010.82 | 181.36 | −136.59 | −8.01 |
| Water Bodies | −46.79 | −39.24 | 1332.09 | 1838.89 | −993.70 | −70.75 | 291.60 | 244.57 |
| Constrution Land | 0.98 | 7.071 | 0.45 | 3.03 | 2.42 | 15.86 | 3.85 | 27.81 |
| Unused Land | 117.39 | 1.19 | −228.15 | −2.29 | −2.36 | −0.024 | −113.12 | −1.15 |

As shown in Tables 3 and 4, areas of unused land decreased by 1.15%, so that the range was not obvious. However, the proportion of unused land was always above 70% from 1990 to 2016. The lower reaches of the Tarim River lie within the Taklimakan Desert. Overall, the largest increase occurred in the water bodies (244.57%), which also confirms the significant effect of the ecological water conveyance. The change rate of water bodies in the period 2000–2010 was very high, whereas a drastic decrease was observed in the following period. The reason for this result was that ecological water conveyance began in 2000, and water resources were "starting from nothing", with the area of water expanding rapidly. Therefore, the change rate in the area of water bodies was greatest from 2000 to 2010. From 2010 to 2016, although the water area was large in 2013, the amount of water transported decreased, leading to a significant decrease in the water area in 2016 compared to that in 2010. As a result, the area of water bodies decreased greatly from 2010 to 2016. In addition, the area of farmland and construction land increased by 51.31% and 27.81% respectively, indicating that human activities strengthened gradually. The area of woodland and grassland decreased by 14.26% and 8.01%, respectively.

### 3.2. Changes in the Ecosystem Service Value Before and After Ecological Water Conveyance

Ecosystem service value equivalents of the unit area in the different years were determined (Table 5) on the basis of Equation (5) and as a result of the application of the work of Constanza et al., according to data from the "Xinjiang Statistical Yearbook" and relevant information from the 'Yearbook of the National Bureau of Statistics'.

**Table 5.** Ecological service value equivalent of unit area of different types of land.

| Year | Types of Land-Use | | | | | |
| | Farmland | Woodland | Grassland | Water Bodies | Construction Land | Unused Land |
| --- | --- | --- | --- | --- | --- | --- |
| 1990 | 0.72 | 12.00 | 2.79 | 28.98 | 13.22 | 0 |
| 2000 | 0.70 | 11.11 | 2.60 | 26.77 | 12.11 | 0 |
| 2010 | 0.87 | 12.98 | 3.07 | 31.18 | 13.95 | 0 |
| 2016 | 0.94 | 13.99 | 3.56 | 33.41 | 15.62 | 0 |

Service value equivalents of the different types of land showed an increasing trend from 1990 to 2016 except for unused land, indicating that ecological water conveyance promoted an increase in the ecological service value per unit area. Based on the service value equivalents, and the area of different land in 1990, 2000, 2010, and 2016, the ecological service value of each land type was calculated, and the changing characteristics that affected the ecosystem service value of each land type were obtained (Table 6).

The value of unused land has always been 0. From 1990 to 2000, in addition to that of farmland, the ecosystem service values of other types of land-use declined. In this period, ecological water conveyance was conducted, so the area of natural vegetation and water decreased. In contrast, human activities have been enhanced, the land reclamation area has increased, and the ecosystem service of farmland has been improved. In 2000, an ecological water conveyance project was put into effect. From 2000 to 2010, the increase in the ecosystem service value of water bodies was the highest. For the area of the desert riparian forest increased, the service value of the woodland increased, human activities increased gradually, and the service value of farmland and construction land also increased. However, the service value of the grassland decreased, mainly because of the decrease in grassland area due to an increase in overgrazing. From 2010 to 2016, the service value of the water bodies declined seriously due to the decrease in the water delivery amount. The service value of other types of land-use increased. In general, from 1990 to 2016, increases in the amplitude of the service value of the water bodies was the highest (297.25%), and the next was farmland (98.76%) and construction land (51.37%), indicating that the ecological water conveyance project has increased the area of the water bodies significantly,

and human activities also enhanced gradually. The increase in the amplitude of the grassland service value was less (17.60%), and the change in the woodland service value was not obvious (–0.01%).

**Table 6.** Change of ecological service value of different land types ($10^8$ dollars).

| Type of Land-Use | 1990–2000 | | 2000–2010 | | 2010–2016 | | 1990–2016 | |
|---|---|---|---|---|---|---|---|---|
| | Variation of Value ($10^8$ Yuan) | Change Rate (%) | Variation of Value ($10^8$ Yuan) | Change Rate (%) | Variation of Value ($10^8$ Yuan) | Change Rate (%) | Variation of Value ($10^8$ Yuan) | Change Rate (%) |
| Farmland | 0.008 | 2.48 | 0.21 | 61.29 | 0.11 | 20.25 | 0.33 | 98.76 |
| Woodland | −3.16 | −12.95 | 2.62 | 12.32 | 0.54 | 2.26 | −0.003 | −0.01 |
| Grassland | −0.43 | −6.70 | −3.71 | −61.40 | 5.28 | 226.53 | 1.14 | 17.60 |
| Water bodies | −2.07 | −43.88 | 57.02 | 2158.33 | −40.96 | −68.66 | 13.99 | 297.25 |
| Construction land | −0.004 | −1.64 | 0.045 | 18.33 | 0.09 | 30.05 | 0.13 | 51.37 |
| Unused land | 0.008 | 0 | 0 | 0 | 0 | 0 | 0 | 0 |

We regard the lower reaches of the Tarim River as a whole and obtained the changing characteristics of the values of each service function for the different years (Table 7).

**Table 7.** Changes in the values of each service function before and after ecological water conveyance.

| Service Function | Total Value of Ecological Service Value ($10^8$ Dollar) | | | | Change Rate (%) | | | |
|---|---|---|---|---|---|---|---|---|
| | 1990 | 2000 | 2010 | 2016 | 1990–2000 | 2000–2010 | 2010–2016 | 1990–2016 |
| Food Production | 0.97 | 1.06 | 3.60 | 2.48 | 9.87 | 239.15 | −31.25 | 156.42 |
| Raw Material Production | 0.93 | 1.01 | 2.51 | 1.74 | 8.37 | 149.92 | −30.69 | 87.81 |
| Gas regulation | 0.55 | 0.39 | 3.95 | 1.61 | −29.21 | 911.98 | −59.24 | 192.33 |
| Climate Regulation | 4.24 | 3.64 | 4.13 | 4.16 | −14.18 | 13.55 | 0.73 | −1.83 |
| Hydrologic Regulation | 0.30 | 0.25 | 0.73 | 0.49 | −16.82 | 190.49 | −31.83 | 65 |
| Waste Disposal | 5.69 | 5.07 | 3.77 | 6.50 | −10.96 | −25.55 | 72.35 | 14.24 |
| Soil Conservation | 14.82 | 12.78 | 13.42 | 13.90 | −13.73 | 4.96 | 3.57 | −6.22 |
| Maintenance of Biodiversity | 3.12 | 2.57 | 10.26 | 5.33 | −17.45 | 298.84 | −48.07 | 71.01 |
| Provide Aesthetic Landscape | 5.57 | 3.76 | 44.34 | 15.72 | −32.60 | 1079.72 | −64.54 | 181.96 |
| Total | 36.18 | 30.52 | 86.70 | 51.76 | −15.63 | 184.04 | −40.30 | 43.06 |

From 1990 to 2016, generally speaking, the ecosystem services values of the lower reaches of the Tarim River showed an increasing trend, and the range of the increase was 43.06%. However, the service values of climate regulation and soil conservation decreased, with reductions in the range of 1.83% and 6.22%, respectively. These reductions were related to the decrease in the area of woodland and grassland. The service value of other functions all increased. The increase in the range of service values for gas regulation, aesthetic landscaping, and food production was 192.33%, 181.96%, and 156.42%, respectively. Furthermore, an increase in the range of service value for raw material production, maintenance of biodiversity and hydrologic regulation were 87.81%, 71.01%, and 65%, respectively. The increase in the range of service value of waste disposal was only 14.24%.

*3.3. Sensitivity Analysis of Ecosystem Services*

Based on the formula used to calculate the sensitivity index, the ecological value coefficients of various land-use types were adjusted up and down 50%, then, the value coefficients after the adjustment were used to calculate the sensitivity indexes of the ecological service value of each land type (Figure 3).

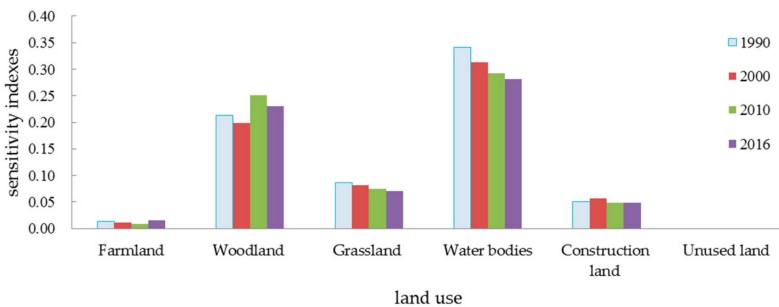

**Figure 3.** Sensitivity index of ecological service value of each type of land in the lower reaches of the Tarim River.

The sensitivity index of the ecosystem service value of each type of land was less than 0.35 after adjusting the value coefficients of each type of land up and down to 50% in 1990, 2000, 2010, and 2016, which shows that the total value of the ecological system was the lack of flexibility in the value coefficients. The results were credible.

### 3.4. Characteristics of Ecological Environment in the Lower Reaches of the Tarim River

According to the standard of "Technical specification for ecological environment assessment (Trial Implementation)", we obtained an index of the ecological environment in the lower reaches of the Tarim River in 1990, 2000, 2010, and 2016 (Figure 4). The environmental condition index decreased by 6.05% from 1990 to 2000, showing that the ecological environment was deteriorating. The environmental condition index increased by 16.51% from 2000 to 2010, showing that the ecological environment was improving gradually, and the environmental condition index increased by 17.81% from 2010 to 2016, indicating that the ecological environment continues to develop in a good direction.

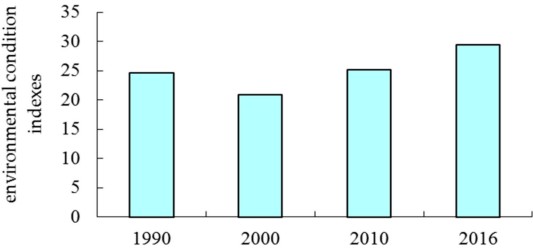

**Figure 4.** Environmental condition index of the lower Tarim River from 1990 to 2016.

In 2016, the environmental condition index was 29.4. Based on the standard grading of the ecological environment (Table 8), the ecological environment in the lower reaches of the Tarim River was at a poor level in 2016.

**Table 8.** Classification of ecological environment (Du et al. 2015).

| Grade | Excellent | Good | Commonly | Poor | Very Poor |
|---|---|---|---|---|---|
| Range of EI | EI ≥ 75 | 55 ≤ EI < 75 | 35 ≤ EI < 55 | 20 ≤ EI < 35 | EI < 20 |

## 4. Discussion

### 4.1. Reasons for a Change in Ecosystem Service Value in the Lower Reaches of the Tarim River

Before the implementation of the ecological water conveyance project, the irrational utilization of water resources and climate such as drought led to a shortage in the water resources in the lower reaches of the Tarim River. Vegetation in the "green corridor" declined, and the ecological environment was deteriorating day by day. It threatened the ecological safety of the river basin and the sustainable

development of the social economy [52]. In 1990, the ecosystem service value was $36.18 \times 10^8$ dollars at a low level. Due to the ecological water conveyance, many changes occurred in the area in terms of the land-use type. Areas of farmland, construction, and water bodies increased heavily due to the ecological water conveyance and increases in the range of water body surface area were the highest, with areas of woodland and grassland declining, though not by much. By 2016, the ecosystem service value had increased to $51.76 \times 10^8$ dollars. This finding shows the benefits of the ecological water conveyance project, which was very effective, and the results are consistent with those of Chen et al. [53], Xu et al. [54], and Ling et al. [55].

Based on these results, the service value equivalents of the various types of land declined from 1990 to 2000. From 1990 to 2000, ecological water conveyance had not yet occurred, the ecological environment continued to deteriorate, people's living standards were low, and the ability of and willingness to pay was poor. From 2000 to 2016, service value equivalents were all on the rise. During this period, because of the impact of ecological water conveyance, the local economy developed rapidly, people's living standards improved, and their ability to pay and willingness to pay increased. Therefore, we should continue to implement the ecological water conveyance rationally in the future and strengthen the protection of woodland, grassland, and other vegetation resources. Moreover, we should develop cultivated land resources rationally. These measures could contribute to the improvement of value equivalence and the total value of the ecosystem services.

### 4.2. Measures for Environmental Quality Restoration in the Lower Reaches of the Tarim River

The ecological environment of the lower reaches of the Tarim River deteriorated from 1990 to 2000 and improved from 2000 to 2016, which was consistent with the results of Huang et al. [56]. Due to natural and human–caused disturbances, the lower reaches of Tarim River had been damaged seriously in the few decades preceding 2000 [57]. In 2000, an ecological water conveyance project began to be implemented, and some protective measures had been taken, with some results achieved, such as the uplift of groundwater level and an increase in the biological abundance index and in the vegetation coverage and water network density index. However, the ecological environment index of the lower reaches of the Tarim River was only 29.4 until 2016. This area was in a poor state, and as suggested by the low index, the ecological problems were still serious. The continued implementation of ecological water conveyance projects is necessary. These projects should be coupled with reasonable ecological regulation measures to promote better ecological restoration and environmental protection in the lower reaches of the Tarim River. A direct relationship exists between the environmental quality and the distribution of vegetation, as well as with the area of water bodies. As a result, the continuation of the ecological water conveyance is important to improve the density and coverage of the vegetation and expand the area of the water bodies. For example, the time frame of ecological water conveyance in a year should form an agreement with the regeneration time of naturally occurring plants [58], so the ecological water transfer should be implemented from August to September every year. At the same time, the germination time of most herbaceous plant seeds is usually in the early spring, so water transport should be considered for implementation in April–May of every year. In this period, the surface evaporation is not great, and the discharge of surface water can also contribute to recharging the groundwater, improving the growth of natural vegetation. For similar inland river basins, water conveyance patterns should be adjusted to match the characteristics of the vegetation, hydrology, and climate, thereby maximizing the benefits of water resource utilization and ensuring the sustainability of the ecosystem in the river basin.

After the implementation of the ecological water conveyance project, the area of water increased, with the beautification of the ecological environment and improvement in the living conditions of the local residents. The area of farmland and construction land increased, indicating that human activities increased gradually, and the intensity of land development and utilization also increased. However, farmland and construction land were reclaimed from woodland and grassland mainly, which resulted in a continuous reduction in these two land-use types [59]. With the increase in farmland,

the amount of water used for agriculture also increased. Ecological water was taken, causing the degradation of natural vegetation, and the ability to resist natural disasters such as desertification was also reduced. The increase in farmland and construction increased the income of residents and promoted local economic development in a short period of time. However, if expansion is unlimited, ecological security will be threatened by hidden dangers. Therefore, in the future, policymakers should follow principles of ecological protection and strengthen the management of land resources, improve the status of land-use, strictly control the growth of farmland and construction, and reduce the damage to natural vegetation, such as woodland and grassland, to ensure ecological resources and benefit the society. As a result, decisionmakers for land-use should carry out alternative protection activities and formulate sound management measures to achieve a balance between sustainable development and ecological protection.

### 4.3. Some Recommendations for Future Research

In this study, the estimation method of ecosystem service value was based on the method proposed by Costanza et al. [13], Xie et al. [47], Shi [48], Shi et al. [49], and Shi et al. [50]. However, ecosystems are characterized by complexity, dynamism, and nonlinearity, and methods of economic assessment were also provided with limitations in different regions [60]. Hence, how to adjust the method to a small scale such as the county and reflect the relationship between economic development of the county and the changes in the ecosystem served as the starting point for this study. Influencing factors of ecosystem service value, which varied with the change in natural and economic conditions, including climate, vegetation, market prices, and trade, etc. [61,62]. This study has solved two main problems. First, in terms of space, a regional adjustment has been applied to the global equivalent factor table to make it suitable for the assessment of ecosystem service value in a small–scale region, and second, time, social payment ability and willingness to pay, which represent a change in economic conditions, were used to economically adjust the equivalent factor to better reflect the dynamic change in ecological service functions over time. As a result, in the study of other similar areas, the above two problems should be considered first, and the value coefficients should be adjusted to conform to the actual situation of the selected research area so that the ecological service value of the land-use in the region can be calculated. Therefore, the proposed methodology in this article can be extended to other contexts.

The vegetation in each land type is not distributed evenly [63], so the accuracy of the value coefficient selected in this study was questionable. For example, some scattered vegetation is also growing on unused land, but the value coefficient of the unused land was assigned to 0. In addition, in the process of calculation, the negative effects of air pollution and water pollution, which may produce negative values, were also neglected [64]. All of the above problems will cause some uncertainties when the ecosystem service values are calculated. In contrast, sensitivity analysis shows that the ecosystem service value was inelastic relative to the value coefficients, indicating that, although the value coefficient is uncertain, the estimation results were stable to a certain extent. More accurate value coefficients should be selected to ensure more reasonable estimation results in future research.

In the process of assessing ecosystem service value, the resolution ratio of the image will also have a great impact on the results [65]. Service values and the changing law of the ecosystem may be different due to different resolution scales. For an image with precise resolution ratio, on the one hand, data will be concentrated on the individual tree rather than the entire woodland, and the estimated value of the forest ecosystem services will be low in such circumstances. On the other hand, the area of the rivers, wetlands and other water bodies will expand, promoting an increase in ecosystem service values. We should focus on resolving the problem of the image resolution ratio and land-use area in future research, especially in towns and rural areas with more land types. This study is not perfect in methodology, but an assessment of ecosystem service values can emphasize sustainable management of ecosystems and support decision making for the sustainable development of the region.

The sum of the GDP (gross domestic product) in Yuli County was $7.51 \times 10^8$ dollars in 2016, and the ecosystem service value of the lower reaches of the Tarim River was $51.76 \times 10^8$ dollars. Therefore,

the value of the natural ecosystem services was 7 times more than that of the regional GDP, which was very different from the result of Chen and Zhang [66] who found that 'China's ecosystem service value was 1.25 times than the GDP'. The main reason for this difference is that the lower reaches of the Tarim River have smaller populations and fewer agricultural, industrial, and tertiary industrial activities, thereby resulting in a lower GDP, which is slightly below the national average level. However, along the lower reaches of the Tarim River, many desert riparian forests, reservoirs and lakes had higher service values than the regional GDP, and the service values of these ecosystems are high. Therefore, a single accounting system such as GDP is not enough to show the ecological and economic conditions. Such a system may lead to an economic falsehood and have adverse effects on long–term ecological construction and environmental restoration projects. Therefore, calculations of GEP (gross ecosystem product) should be implemented to improve the scientific estimation of regional ecological values.

This article provides a case study for evaluating service values of natural ecosystems. However, compared with natural ecosystems, urban ecosystems are much more complicated [67]. In the past, many studies have focused on natural ecosystems, but future research on ecosystem service value assessments should pay more attention to the ecosystem in urban areas, which will show strong interactions between humans and the ecosystem, which will enable assessment techniques to guide human activities in the future.

## 5. Conclusions

To understand the response of ecosystem service values and environmental quality following ecological water conveyance to downstream areas in arid inland river basins, the Tarim River, which travels along a typical inland river basin, was selected as the study area, and the effects of changing characteristics on ecosystem service values and environmental quality in the basin were analysed.

From 1990 to 2016, the area of unused land decreased by 1.15%, and areas of water bodies increased by 244.57%. The area of farmland and construction land increased by 51.31% and 27.81%, respectively, and the area of woodland and grassland decreased by 14.26% and 8.01%, respectively. In terms of the ecological service value of all types of land, the water bodies with increased area showed the greatest increase in value (297.25%), with farmland and construction being next. An increase in the amplitude of the service value of grassland was less (17.60%) and that of woodland was not obvious (–0.01%). From 2000 to 2016, the environmental condition index showed a general increase. In 2016, the environmental condition index was 29.4, and the ecological environment of the lower reaches of the Tarim River was at a poor level in 2016. Ecosystem service values and ecological environmental quality are closely related to the vegetation and water area in the lower reaches of the Tarim River.

In the process of ecological water conveyance, farmland and construction land have increased, but the increase in the area of the water bodies has increased the most. Furthermore, the ecological service value per unit area of water bodies is much higher than that of farmland and construction land. Therefore, water bodies have contributed significantly to the increase in the ecosystem service value. In addition, water also accounts for a large proportion in the calculation formula of the environmental condition index, which involves the river length/area, lake (offshore) length/area, water resource amount, etc. Therefore, we can conclude that the transport of ecological water brought great benefits.

Ecological water conveyance has beautified the environment and has promoted economic development in the lower reaches of the Tarim River. The area of cultivated land and construction land has increased, but these increases are mainly due to the reclamation of woodlands and grasslands. In addition, an increase in irrigation water usage has consumed much of the ecological water. Therefore, ecological restoration is not sustainable in the basin. In the future, priority should be given to ecological protection from the beginning, which should be followed by controlling the amount of land reclamation and by increasing the yield per unit area. Woodlands, grasslands and other ecological resources, therefore, should be protected, and the sustainable balance between ecological protection and economic development should be determined.

**Author Contributions:** Conceptualization and methodology, X.W. and H.L.; writing—original draft preparation, X.W.; supervision and data analysis, X.W. and S.P.; project administration and funding, H.X. and H.L.; data processing, X.W. and T.M.

**Funding:** This work was supported by the National Natural Science Foundation of China (31360101) and the Youth Innovation Promotion Association Project (CAS).

**Acknowledgments:** The authors would like to express their cordial gratitude for assistance in this research to XinFeng Zhao, GuangPeng Zhang and Le Zhao in Xinjiang Institute of Ecology and Geography, Chinese Academy of Sciences, Urumqi, China.

**Conflicts of Interest:** The authors declare that there is no conflict of interests regarding the publication of this paper.

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
