# Peer review of "Do Ecosystem Service Value Increase and Environmental Quality Improve due to Large–Scale Ecological Water Conveyance in an Arid Region of China?"

_sustainability, doi:10.3390/su11236586_

Round 1

Reviewer 1 Report

The proposed study is aimed at assessing the changes in the land use and ecosystem service value of the lower reaches of the Tarim river during the last 30 years, after the investment of the Chinese government to build a large scale water conveyance system.

The paper is interesting and the topic is adequate for Sustainability MDPI.

However, the manuscript is mainly related to the case studio of the lower reaches of the Tarim River, which has local validity and cannot be easily transferred to other areas. Moreover, according to the authors, the study is imperfect (line 433) and there are a certain bias and some uncertainties in the applied methodology (line 414). The Authors should emphasize if and how the proposed methodology can be extended under other contexts.

The paper is poorly written and some sentences appear not clear, due to the poor language. Other general corrections are following indicated, referring to the pdf file for further comments and suggestions.

Line 38 and others. Acronyms should be defined when they are introduced in the text.

The quality of the figures is quite low. It is not possible to see the places indicated in the text (lines 105-108).

Line 126. It seems that Authors have elaborated only 4 images, one per year. If so, why they did not consider the image retrieved in close days of the different years? Why did they consider the periods from July to September?

The caption of figure 2 should be changed in “Land use in the lower reaches of the Tarim River in four years from 1990 to 2016.

After the equations, it should be used in the present tense to illustrate the meaning of the different symbols.

I suggest to avoid subsections inside the paragraphs, as well as to avoid the use of numbers inside the circle as bullets points.

Paragraph 3.1. Authors should provide the total extension of the investigated area in km2, to assess the amplitude of the observed changes. Even in figure 3 more than the proportion, it should be indicated the absolute values associated with each type of land area.

Table 3 does not report the area associated with unused land. Is there any explanation because the change rate of water bodies in the period 2000-2010 is so high whereas in the following period is drastically decreasing?

Figure 4 and Table 5 show the same data. I suggest leaving only the table, deleting figure 4.

Punctuation throughout the text has to be revised and the capital letter after “;” corrected, because they are not necessary.

The authors should extensively edit the English language, by referring to a native speaker.

Author Response

Response to Reviewer 1 Comments

One thing we want to get across is that the article “Does the Value of Eco-service Increase and the Environmental Quality Improve due to Large-scale Ecological Water Conveyance in an Arid Region of China?(Manuscript ID sustainability-622832)” was sent to an English editing company to improve the English language. In view of your requirements, we have also urged the company several times, but it has not been finished. Therefore, we can only send you the revised article without perfect language. Please review it firstly. After the language improved, we will return the revision as soon as possible. Thank you!

Comments and Suggestions for Authors:

The proposed study is aimed at assessing the changes in the land use and ecosystem service value of the lower reaches of the Tarim river during the last 30 years, after the investment of the Chinese government to build a large scale water conveyance system.

The paper is interesting and the topic is adequate for Sustainability MDPI.

Point 1: However, the manuscript is mainly related to the case studio of the lower reaches of the Tarim River, which has local validity and cannot be easily transferred to other areas. Moreover, according to the authors, the study is imperfect (line 433) and there are a certain bias and some uncertainties in the applied methodology (line 414). The Authors should emphasize if and how the proposed methodology can be extended under other contexts.

Response 1: Thanks for your point. We conclude that the proposed methodology can be extended under other contexts. We discuss it in “4.3 Some recommendations for future research” (Line 439-456):

“In this study, the estimation method of ecosystem service value was based on the method proposed by Costanza et al [13], Xie et al.[47], Shi [48], , Shi et al.[49], Shi et al.[50]. However, ecosystem is characterized by complexity, dynamism and nonlinearity, and methods of economic assessment were also provided with limitations in different region [60]. Hence, how to adjust it to a small scale of county and reflect the relationship between economic development of county and changes of ecosystem is the starting point of this study. Influencing factors of ecosystem service value, which varies with the change of natural and economic conditions, include climate, vegetation, market payment and trade, etc. [61,62]. This study has solved two main problems: In terms of space, regional adjustment has been applied to the global equivalent factor table in order to make it suitable for the assessment of ecosystem service value in a small- scale region; in terms of time, social payment ability and willingness to pay, which represent the change of economic conditions, were used to economically adjust the equivalent factor to better reflect the dynamic change of ecological service function over time. As a result, in the study of other similar areas, the above two problems should be considered firstly, that value coefficients should be adjusted to conform to the actual situation of the selected research area; secondly, we can calculate the ecological service value of land use in this region. Therefore, proposed methodology in this article can be extended under other contexts.”

.

”.

Point 2: The paper is poorly written and some sentences appear not clear, due to the poor language. Other general corrections are following indicated, referring to the pdf file for further comments and suggestions.

Response 2: Thanks for your point. We asked an English editor to polish the article to meet the requirements of the journal. In view of your requirements, we have also urged the company several times, but it has not been finished. Therefore, we can only send you the revised article without perfect language. Please review it firstly. After the language improved, we will return the revision as soon as possible. We have modified the paper referring to the comments in the PDF file, and all the modified parts are marked in red in the paper.

Point 3: Line 38 and others. Acronyms should be defined when they are introduced in the text.

Response 3: Thanks for your point. We have defined all of the acronyms. For example, IHDP (International Human Dimensions Programme on Global Environmental Change); GEP (gross ecosystem product); TM (Thematic Mapper) ; CBERS (China-Brazil Earth Resource Satellite); COD (Chemical Oxygen Demand); GDP (Gross Domestic Product).

Point 4: The quality of the figures is quite low. It is not possible to see the places indicated in the text (lines 105-108).

Response 4: Thanks for your point. We have modified the figures to improve the quality.

Point 5: Line 126. It seems that Authors have elaborated only 4 images, one per year. If so, why they did not consider the image retrieved in close days of the different years? Why did they consider the periods from July to September?

Response 5: Thanks for your point. After reviewing the data, we found that the imaging time of these 4 images was August in each year respectively. We added the following in the article (Line 149-151):

“It was a period of abundant water in the basin, with large amount of water in rivers and reservoirs; What’s more, natural vegetation and crops grow luxuriantly, and land features were obvious”.

Point 6: The caption of figure 2 should be changed in “Land use in the lower reaches of the Tarim River in four years from 1990 to 2016.

Response 6: Thanks for your point. We have changed the caption of figure 2 to “Land use in the lower reaches of the Tarim River in four years from 1990 to 2016”. (Line 163)

Point 7: After the equations, it should be used in the present tense to illustrate the meaning of the different symbols.

Response 7: Thanks for your point. We made some changes. In the article, we illustrate the meaning of the different symbols use in the present tense.  

Point 8: I suggest to avoid subsections inside the paragraphs, as well as to avoid the use of numbers inside the circle as bullets points.

Response 8: Thanks for your point. We have wiped out the subsections. Such as:

“(1) Choice and correction of value coefficients. (2) Calculation of ecosystem service value. (3) Sensitivity analysis

We also removed the numbers inside the circle as bullets points. (Line 250-266)

Point 9: Paragraph 3.1. Authors should provide the total extension of the investigated area in km2, to assess the amplitude of the observed changes. Even in figure 3 more than the proportion, it should be indicated the absolute values associated with each type of land area.

Response 9: Thanks for your point. First, let's explain. In this study, lower reaches of Tarim River was regarded as a whole and the land was divided into six categories. Changing of land is the mutual transformation among these kinds of land, and the total area of the land will not change. Therefore, this study cannot provide the change characteristics of the total extension of the investigated area. Secondly, in the article, we added Table 3, and removed Figure 3, thus we indicated the absolute values associated with each type of land area. (Line 280-281.)

Table 3. Areas of different types of land area in different years

Type of land use

1990

2000

2010

2016

Farmland

337.39

355.02

458.38

510.5041

Woodland

1493.93

1404.67

1350.29

1280.89

Grassland

1704.77

1704.84

557.36

1568.18

Water bodies

119.23

72.44

1404.53

410.83

Constrution land

13.86

14.84

15.29

17.71

Unused land

9828.96

9946.35

9718.2

9715.84

Point 10: Table 3 does not report the area associated with unused land. Is there any explanation because the change rate of water bodies in the period 2000-2010 is so high whereas in the following period is drastically decreasing?

Response 10: Thanks for your point. We added the area associated with unused land (Line 283-285):

“As shown in Table 3 and Table 4, area of unused land decreased by 1.15%, that the range was not obvious. However, the proportion of unused land has always been above 70% from 1990 to 2016.”.

According to the change characteristics of water body area, the following contents are added in the article (Line 287-295):

“The change rate of water bodies in the period 2000-2010 is so high whereas in the following period is drastically decreasing. The reason was that the ecological water conveyance began in 2000, and water resources have experienced the process of "starting from nothing" and the water area has expanded rapidly. Therefore, the change rate of water bodies area is great from 2000 to 2010. From 2010 to 2016, although the water area was large in 2013, the amount of water transported has decreased since then, leading to a decrease in the water area in 2016, which is significantly lower than the water area in 2010. As a result, the area of water bodies decreased greatly from 2010 to 2016”.

Point 11: Figure 4 and Table 5 show the same data. I suggest leaving only the table, deleting figure 4.

Response 11: Thanks for your point. According to your suggestion, we deleted the figure.

Point 12: Punctuation throughout the text has to be revised and the capital letter after “;” corrected, because they are not necessary.

Response 12: Thanks for your point. We reread the text and revised the inappropriate punctuation. In addition, we corrected the capital letters after “;”.

Point 13: The authors should extensively edit the English language, by referring to a native speaker.

Response 13: Thanks for your point. We asked an English editing company to polish the article to meet the requirements of the journal. In view of your requirements, we have also urged the company several times, but it has not been finished. Therefore, we can only send you the revised article without perfect language. Please review it firstly. After the language improved, we will return the revision as soon as possible.

Reviewer 2 Report

Keywords: Replace “lower reaches of Tarim river” by “Tarim river” and add “arid area”.

The summary needs to be improved. The author should explain the objectives of the study and briefly indicate the main results.

Figures have poor graphic quality and no reading. They must be improved.

1.Introduction

The author should present bibliographical research on the concept of ecological water conveyance and present examples of other case studies.

Line 36 - indicate the meaning of the acronym IHDP.

Data and Methods

In the equations indicate the units of the variables. The units must be in the international system (SI). Review the units throughout the article.

Lines 111/112 - briefly explain what the ecological water conveyance project (2000) is and indicate the bibliographic reference.

Line 122 - indicate the meaning of the acronym CMB.

Line 155 - Is the desert system the same of the unused land (Table1)?

The author must present the data concerning the ecological transport of water.

Results and analysis

Line 237 - Table 2 - indicate the meaning of COD.

Line 262 - Table 4 - The digits should not have as many decimal places Chinese yuan should not be used as currency (eg lines 195, 198, 272).

The author should explain why he considers The value of unused land has always been 0 (Table 4, Line 273).

Figure 5 - YY axis: missing legend and all digits must be two decimal places.

Line 312 - for readability put the acronyms in full.

Figure 6 - Erase Age on the XX axis. It is Ecological environmental index or Environmental condition index (Line 322). Standardize the language in the text.

Discussion

Lines 350-355 - The author should explain more clearly how the study solves two problems.

Line 412 - The bibliographic reference is missing. Not enough Costanza et al.

Line 436 - indicate the meaning of acronym GDP.

Conclusion

The author reaches conclusions that do not reflect the results of this work.

It is not clear how it can be concluded that the benefits were achieved due to the ecological transport of water in the lower regions of the Tarim River.

The author must demonstrate how he has come to the conclusion that the best ecological water transport period is April-May and August-September of each year.

Author Response

Response to Reviewer 2 Comments

One thing we want to get across is that the article “Does the Value of Eco-service Increase and the Environmental Quality Improve due to Large-scale Ecological Water Conveyance in an Arid Region of China?(Manuscript ID sustainability-622832)” was sent to an English editing company to improve the English language. In view of your requirements, we have also urged the company several times, but it has not been finished. Therefore, we can only send you the revised article without perfect language. Please review it firstly. After the language improved, we will return the revision as soon as possible. Thank you!

Comments and Suggestions for Authors

Point 1: Keywords: Replace “lower reaches of Tarim river” by “Tarim river” and add “arid area”.

Response 1: Thanks for your point. In this article, lower reaches of Tarim river was taken as the research object, and the ecological environment of the middle and upper reaches were obviously different from that of the lower reaches. Therefore, the key words of this article are still used “lower reaches of Tarim river”. However, we also accept your another suggestion, that add “arid area”. (Line 35)

Point 2: The summary needs to be improved. The author should explain the objectives of the study and briefly indicate the main results.

Response 2: Thanks for your point. We add the objectives of the study in the summary:

“The objective of this study is to verify whether the ecological water conveyance project promotes the increase of ecosystem service value, so as the improvement of ecological environment, that in order to provide references for the effect evaluation of ecological water conveyance, so as the management of water resources. In this way, economic development can be coordinated with environmental protection. Thus, The economy can develop sustainably”. (Line 19-24)

We also briefly indicate the main results:

“from 1990 to 2016, ecosystem service value has been substantially rise, indicating benefits of ecological water conveyance were very significantly. Environmental condition index increased by 21.14%, shows that ecological environment has improved”. (Line 27-30).

Point 3: Figures have poor graphic quality and no reading. They must be improved.

Response 3: Thanks for your point. We have improved the quality of Figures.

Introduction

Point 4: The author should present bibliographical research on the concept of ecological water conveyance and present examples of other case studies.

Response 4: Thanks for your point. We have added the bibliographical research on the concept of ecological water conveyance and examples of other case studies (Line 82-91):

At present, there are some researches on concept of ecological water conveyance. Some experts believe ecological water conveyance is a water conveyance project to protect and restore the ecosystem with natural vegetation [28]. In addition, some experts believe that ecological water conveyance is a project to balance water resources without destroying the ecological environment, so that the project is integrated with nature [29]. At present, most of the researches on ecological water conveyance are focused on China's arid areas, especially Teihe and Tarim river basins [30, 31]. The research on ecological water conveyance in Heihe river basin mainly focuses on the influence of water conveyance on vegetation [32, 33], the characteristics of water resources in the basin [34], and the comprehensive management of water-eco-economic system [35].

Point 5: Line 36 - indicate the meaning of the acronym IHDP.

Response 5: Thanks for your point. We have indicated the meaning of the acronym IHDP:

“"IHDP (International Human Dimensions Programme on Global Environmental Change) " (Line 37-38)

Data and Methods

Point 6: In the equations indicate the units of the variables. The units must be in the international system (SI). Review the units throughout the article.

Response 6: Thanks for your point. We have added the units of the variables. However, some variables were not with units, such as the coefficients including l, e, T etc., that we haven’t added the units. The revised content has been marked in red in the article.

Point 7: Lines 111/112 - briefly explain what the ecological water conveyance project (2000) is and indicate the bibliographic reference.

Response 7: Thanks for your point. We have briefly explain what the ecological water conveyance project (2000) is and indicate the bibliographic reference:

“In order to save the strategic “green corridor” in the lower reaches of Tarim river, an emergency ecological water conveyance project in the lower reaches of tarim river was started in May 2000. Aim of the project is to raise the water level on both sides of the river and save the declining natural vegetation [44, 45].” (Line 123-126.)

Point 8: Line 122 - indicate the meaning of the acronym CMB.

Response 8: Thanks for your point. We redescribed the source of the data:

“The basic data were Landsat-TM (Thematic Mapper) image data and CBERS (China-Brazil Earth Resource Satellite) data in the lower reaches of Tarim River”. (Line 147-148.)

Point 9: Line 155 - Is the desert system the same of the unused land (Table1)?

Response 9: Thanks for your point. We have explained it in the article. (Line 180-188)

“In arid areas, water is an important factor that determines the characteristics of ecosystem, and also determines the two ecological environment evolution processes of oasization and desertification, which are extremely contradictory and conflicting. The role and value of water should be highlighted in arid areas, thus value system of Constanza et al. is more applicable to the actual situation of arid areas. In addition, value system Constanza et al., unused land includes desert and gobi. Desert and gobi, although the weight of which is not high, but the area is huge, occupy a large proportion in arid areas. Therefore, the value of unused services is 0, which is more in line with the actual situation of arid areas. As a result, this article chooses the value system of Constanza et al. as the basis.”

Point 10: The author must present the data concerning the ecological transport of water.

Response 10: Thanks for your point. We have presented the data concerning the ecological transport of water. (Line 128-137)

“From 2000 to 2016, there were 17 times ecological water transfers carried out in the lower reaches of Tarim river, with a cumulative discharge of 58.14 ×108 m3 and an average annual discharge of 3.42 ×108 m3. The water head reached Taitema lake for 14 times, which effectively replenished the groundwater. With the rise of groundwater level, water quality of groundwater was getting better and the water environment was improved obviously. From 2000 to 2016, the groundwater depth increased from 9.8-10.1 m to 2.1-5.3 m at 1 km away from the river. The area of vegetation restoration and improvement reached 2285 km2, and the area of new vegetation coverage reached 362 km2. Area of sand decreased by 854 km2, and a wetland about 223 km2 was formed around Taitema lake.”

Results and analysis

Point 11: Line 237 - Table 2 - indicate the meaning of COD.

Response 11: Thanks for your point. We have indicated the meaning of COD:

“COD (Chemical Oxygen Demand)” (Line 269, Table 2)

Point 12: Line 262 - Table 4 - The digits should not have as many decimal places Chinese yuan should not be used as currency (eg lines 195, 198, 272).

Response 12: Thanks for your point. We set the digits to hold only two decimal places. What’s more, According to your suggestion: “Chinese yuan should not be used as currency”, we changed the yuan to dollar. This paper chooses the average exchange rate between us dollar and RMB from 2000 to 2016, and the data were recalculated. The revised content has been marked red in the text

Point 13: The author should explain why he considers The value of unused land has always been 0 (Table 4, Line 273).

Response 13: Thanks for your point. We have explained in point 9.

Point 14: Figure 5 - YY axis: missing legend and all digits must be two decimal places.

Response 14: Thanks for your point. We have modified it according to your suggestion (Line 347-348)

Point 15: Line 312 - for readability put the acronyms in full.

Response 15: Thanks for your point. We have added the acronyms, ESV was changed to “the total value of ecological system”, and E was changed to “value coefficients”. (Line 352-353)

Point 16: Figure 6 - Erase Age on the XX axis. It is Ecological environmental index or Environmental condition index (Line 322). Standardize the language in the text.

Response 16: Thanks for your point. We have modified it according to your suggestion, and we define it as the “environmental condition index”.  (Line 357-365)

Discussion

Point 17: Lines 350-355 - The author should explain more clearly how the study solves two problems.

Response 17: Thanks for your point. In this study, we aimed to resolve 2 key scientific issues:

reveal the changing characteristics of land use in the region, and obtain the change rule of ecosystem service value that caused by it. In the discussion, we explained the changing characteristics of land use and ecosystem service value, then analyzed the reasons (Line 375-389):

“In 1990, ecosystem service value was 36.18×108 dollar, at a low level. Because of the ecological water conveyance, there were many changes on the area in terms of land use type. Areas of farmland, construction land and waterbodies increased heavily due to the ecological water conveyance, and increased range of waterbodies surface area was the highest; areas of woodland and grassland declined, but not much. In 2016, ecosystem service value has been increased to 51.76×108 dollar. It shows that benefit of ecological water conveyance project was very effective, that was consistent with the results of Chen et al. [53], Xu et al. [54], Ling et al. [55].

Based on the results, service value equivalence of various types of land declined from 1990 to 2000. From 1990 to 2000, the ecological water conveyance was not started yet, the ecological environment deteriorated continuously, people's living standards were low, and their ability to pay and willingness to pay were poor. From 2000 to 2016, service value equivalent were all on the rise. During this period, because of the impact of ecological water conveyance, the local economy developed rapidly, people's living standard improved, and their ability to pay and willingness to pay increased”.

evaluate whether the environmental quality became better or worse. In the discussion, we also analyze the rule of the change of environmental condition index. (Line 395-403)

Ecological environment in the lower reaches of Tarim River deteriorated from 1990 to 2000 and improved from 2000 to 2016, which was consistent with the results of Huang et al. [56]. Because of disturbances of natural and human-beings, lower reaches of Tarim River had been damaged seriously in the past few decades before 2000 [57]. In 2000, ecological water conveyance project began to be implemented, and some protective measures have been taken and achieved some results, such as the uplift of groundwater level, the increase of biological abundance index, vegetation coverage and water network density index. However, the ecological environment index of the lower reaches of Tarim River was only 29.4 until 2016. It was in a poor state, that indicating the ecological problems were still serious”.

Point 18: Line 412 - The bibliographic reference is missing. Not enough Costanza et al.

Response 18: Thanks for your point. We have added other experts and literature cited in this article. (Line 440)

Point 19: Line 436 - indicate the meaning of acronym GDP.

Response 19: Thanks for your point. We have indicate the meaning of acronym GDP (Gross Domestic Product). (Line 478)

Conclusion

Point 20: The author reaches conclusions that do not reflect the results of this work.

Response 20: Thanks for your point. We rearranged the conclusion to reflect the results of this study. (Line 504-514)

“From 1990 to 2016, area of unused land decreased by 1.15%, areas of water bodies increased 244.57%. The area of farmland and construction land increased by 51.31% and 27.81% respectively, and the area of woodland and grassland decreased by 14.26% and 8.01% respectively. In terms of ecological service value of all types of land, increased range of water bodies was the highest (297.25%), farmland and construction land were the next; Increase amplitude of service value of grassland was less (17.60%), and that of woodland was not obvious (-0.01%). From 2000 to 2016, environmental condition index was increased in general. In 2016, environmental condition index was 29.4,that the ecological environment of lower reaches of Tarim river was at a poor level in 2016. Ecosystem service value and ecological environment quality are closely related to vegetation and water area in the lower reaches of Tarim River.

Point 21: It is not clear how it can be concluded that the benefits were achieved due to the ecological transport of water in the lower regions of the Tarim River.

Response 21: Thanks for your point. We have the following explanations. (Line 515-522)

“In the process of ecological water conveyance, farmland and construction land have increased, but the water bodies has increased the highest; what’s more, the ecological service value per unit area of water bodies is much higher than that of farmland and construction land. Therefore, water bodies have contributed significantly to the increase of ecosystem service value. In addition, water also accounts for a large proportion in the calculation formula of environmental condition index, that involved river length/area, lake (Offshore) area/area , and water resource amount ect. Therefore, we can conclude ecological water transportation has brought great benefits”.

Point 22: The author must demonstrate how he has come to the conclusion that the best ecological water transport period is April-May and August-September of each year.

Response 22: Thanks for your point. “the best ecological water transport period is April-May and August-September of each year.” is part of the discussion section, we have described it in the discussion. (Line 408-416).

Round 2

Reviewer 1 Report

Despite the Authors have satisfactorily answered to all the comments indicated in the first revision, the manuscript still need an extensive revision of English language. Moreover, the table and the text should be re-formatted to be adequate to the Journal standard.

Reviewer 2 Report

No comments.